# Metabolic “Sense Relay” in Stem Cells: A Short But Impactful Life of PAS Kinase Balancing Stem Cell Fates

**DOI:** 10.3390/cells12131751

**Published:** 2023-06-30

**Authors:** Chintan K. Kikani

**Affiliations:** Department of Biology, College of Arts and Sciences, University of Kentucky, Thomas Hunt Morgan Building, 675 Rose Street, Lexington, KY 40506, USA; chintan.kikani@uky.edu

**Keywords:** PAS Kinase, PASK, muscle regeneration, mitochondria, mTOR, myogenesis, Pax7

## Abstract

Tissue regeneration is a complex molecular and biochemical symphony. Signaling pathways establish the rhythmic proliferation and differentiation cadence of participating cells to repair the damaged tissues and repopulate the tissue-resident stem cells. Sensory proteins form a critical bridge between the environment and cellular response machinery, enabling precise spatiotemporal control of stem cell fate. Of many sensory modules found in proteins from prokaryotes to mammals, Per-Arnt-Sim (PAS) domains are one of the most ancient and found in the most diverse physiological context. In metazoa, PAS domains are found in many transcription factors and ion channels; however, PAS domain-containing Kinase (PASK) is the only metazoan kinase where the PAS sensory domain is connected to a signaling kinase domain. PASK is predominantly expressed in undifferentiated, self-renewing embryonic and adult stem cells, and its expression is rapidly lost upon differentiation, resulting in its nearly complete absence from the adult mammalian tissues. Thus, PASK is expressed within a narrow but critical temporal window when stem cell fate is established. In this review, we discuss the emerging insight into the sensory and signaling functions of PASK as an integrator of metabolic and nutrient signaling information that serves to balance self-renewal and differentiation programs during mammalian tissue regeneration.

## 1. Introduction

In electronics, a “sense relay” typically refers to a type of relay that is used to monitor or sense certain parameters or conditions within a circuit. A sense relay is often employed to provide feedback or control based on the status or measurement of the monitored parameter. Thus, sense relays are often integrated into automation control systems or protection schemes to ensure the safe and efficient operation of electrical equipment and systems. Sensory kinases provide similar functional advantages in the cellular context by sampling the environment and relaying appropriate responses for timely decision-making. In stem cells, sensory information is critical to establish lineage identity and self-renewal or differentiation decisions. Thus, protein kinase signaling plays a pivotal role in controlling cell identity, pluripotency, and self-renewal of stem cells [1,2,3]. Stem cells rely on specific soluble factors to maintain a proliferative state that ensures the preservation of their cell identity and differentiation potential, such as the Leukemia Inhibitor Factor (LIF) or basic Fibroblast Growth Factor (bFGF) [4,5]. These factors stimulate the proliferative signaling pathways and inhibit the differentiation to maintain self-renewal. For example, LIF activation of the JAK-STAT signaling pathway plays a crucial role in maintaining the pluripotent state of mESCs. LIF could also activate the PI-3K-AKT pathway in parallel to maintain the expression of pluripotency factors [6,7]. In addition to LIF, Wnt signaling is crucial for the continued self-renewal and pluripotency of mouse embryonic stem cells (mESCs) [8]. Inhibition of GSK-3β is one mechanism for stimulating proliferative response in stem cells in response to Wnt and/or PI-3K/Akt signaling. GSK3β stimulates the degradation of β-Catenin via phosphorylation, a key transcriptional activator of the stem cell proliferation program. Thus, inhibition of GSK-3β by CHIR99021 along with MAP Kinase/ERK Kinase (MEK) inhibitor (PD0325901) is used to maintain the ground state of mouse ESC. Therefore, targeting the protein kinases within these pathways has emerged as a key strategy to maintain the pluripotency of human and mouse embryonic stem cells in vitro. Notably, of 534 human kinases, only a handful of protein kinases have been explicitly studied in the context of stem cell identity, cell-fate decisions, and differentiation. However, for many of these kinases, their stem cell functions are often a biochemical extension of their essential roles in normal physiology in non-stem cells and tissues. Our recent discovery of a stem cell-enriched protein kinase, PASK, opens avenues for investigating the signaling functions of kinases specifically in the stem cell context.

PAS Kinase (PASK) consists of two tandemly arranged Per-Arnt-Sim (PAS) domains, terms PAS-A and PAS-B, respectively, and a catalytic domain within the CAMK family of protein kinases [9,10] (Figure 1). The solution structure of the PAS-A domain has been solved, which revealed a canonical PAS fold consisting of five anti-parallel β-sheets and several α-helices (Figure 1) [11]. Notably, the PAS-A domain structure also revealed structural similarity to the bacterial FixL PAS domain, which contains a flexible regulatory loop region involved in sensory functions. Based on these observations, it is tempting to speculate that the mammalian PAS-A domain could serve a signaling function by binding small metabolites, which in turn could regulate the functions of the cognate catalytic domain. The catalytic domain of PASK is a close structural neighbor of PIM proto-oncogene kinase, displaying a number of structural similarities and closely matching the substrate preference (Figure 1) [12]. PASK is a constitutively active protein kinase that maintains the active structural fold in the absence of activation loop phosphorylation, suggesting that alternative modes of regulation might exist to regulate its function in accordance with cellular needs [12]. One mode of regulation of PASK catalytic activity is hypothesized to occur via intramolecular association of the catalytic domain with the sensory PAS domain [11]. While in vitro studies showed the inhibitory effect of a large excess of purified PAS domain on the catalytic activity of PASK, it remains unclear if the PAS domain inhibits the catalytic domain physiologically [11]. Instead, our recent discoveries reveal that the primary mode to control the PASK functions in stem cells appears to center around acute transcriptional regulation, subcellular distribution, and further catalytic activation by metabolic and nutrient signaling pathways.

## 2. Stem Cell-Specific Expression of PASK

Under normal physiological conditions, PASK mRNA and proteins are nearly absent in adult human and mouse tissues [13,14]. Therefore, we profiled large publicly available mRNA expression datasets spanning multiple tissues and cell types to identify a cell or tissue system where PASK mRNA and protein levels are elevated. Our analyses revealed significant enrichment of PASK mRNA levels in undifferentiated stem and progenitor cell types [13]. Thus, mouse and human embryonic stem cells, cultured myoblasts, pre-adipocytes, and neuronal progenitor cells display increased levels of PASK mRNA. In contrast, the differentiated counterparts of these stem cells, such as embryoid bodies, myotubes, mature adipocytes, and neurons, showed greatly reduced levels of PASK mRNA expression [13]. These results indicated that PASK expression is linked to the undifferentiated state of stem cells.

## 3. ESCs and iPSCs

Analysis of bulk RNAseq datasets revealed a positive correlation between the undifferentiated state of mouse ESCs and the mRNA levels of PASK. In contrast, induction of neurogenesis of mESCs or an embryoid body formation resulted in a progressive decline in PASK expression in tune with differentiation kinetics [13]. Similarly, cellular reprogramming of mouse embryonic fibroblasts (MEFs), in which PASK is nearly absent, into induced pluripotent stem cells (iPSCs) resulted in increased PASK expression [15]. However, when a cardiac-specific differentiation program was initiated, PASK mRNA expression was progressively lost upon differentiation [16]. Thus, PASK expression is tightly linked with the undifferentiated state of stem cells (Figure 2). Examining ChIP-seq data from mESCs or iPSCs, we noticed Oct4 occupancy at the PASK promoter, which correlated with RNA PolII recruitment and histone H3 lysine 4 (H3K4me3) marks, indicative of active promoter in stem cells [17]. Consistent with the notion that Oct4 might be an upstream transcriptional activator of PASK, silencing of Pou5f1 resulted in decreased PASK expression [18].

Despite increased expression during iPSC reprogramming, the catalytic inhibition of PASK did not affect the iPSC reprogramming efficiency, indicating that PASK may not play a significant role during iPSC reprogramming [13]. On the other hand, PASK inhibition nearly completely abolished the terminal differentiation of ESCs into the neuronal lineage, indicating a functional role of PASK during ESC differentiation [13]. Furthermore, PASK inhibition resulted in the maintenance of Oct4 levels despite retinoic acid treatment, indicating sustained preservation of self-renewal by PASK inhibition [13]. This is interesting since kinase inhibitors such as CHIR99021 (GSK3βi) and PD0325901 (ERK1/2i) and in combination with SB203580 (p38i) and SP600125 (JNKi) are used to achieve and maintain ground-state pluripotency of mouse and human ESCs. Our recent studies revealed that PASKi treatment performed at least as well as 2i treatment in maintaining the expression of Oct4, Sox2, and Rex1 in proliferating mESCs, and prevented mESC differentiation upon LIF withdrawal [19]. Intriguingly, the release of ESCs from PASKi resulted in robust embryoid body formation, compared with 2i-treated ESCs, suggesting an improvement in the preservation of ESC stemness by PASKi treatment [19]. These results point to the functional roles of PASK signaling in regulating ESC self-renewal, stemness, and differentiation pathways. However, further studies are required to understand the mechanistic bases of these PASK functions, its molecular targets in ESCs, and explore the functional crosstalk between PASK and other kinases with established stem cells functions, such as GSK3β, ERK1/2, p38MAPK, and JNK.

## 4. Muscle Satellite Cells

Muscle satellite cells, or muscle stem cells (MuSCs), offer a unique biological context to dynamically study the regulation and function of PASK during mammalian tissue regeneration. Uninjured quiescent MuSCs cycle infrequently and are metabolically less active (Figure 3). Damage to the myofiber network due to mechanical or chemical injury results in activation of MuSCs [20,21]. MuSCs activation induces a metabolic shift resulting in increased mitochondrial remodeling, oxidative phosphorylation, and reactive oxygen species generation (Figure 3). These metabolic changes are linked with changes in mitochondrial morphology and network [22,23]. These mitochondrial remodeling fuels biosynthetic pathways leading to increased MuSCs cell size and their cell cycle entry. Quiescent MuSCs express paired-box transcription factor, Pax7. Activation of MuSCs results in translational induction of E-box family protein, MyoD, and Myf5. Pax7, MyoD, and Myf5 are expressed at various levels in activated proliferating myoblasts, resulting in a highly heterogenous myoblast population. Ultimately, transcriptional induction of Myogenin (MyoG) in MyoD positive myoblasts marks onset of terminal differentiation program (Figure 3) [20,21,24,25].

PASK is not expressed in adult skeletal muscles, and its protein levels are absent from quiescent muscle stem cells. However, myofiber injury rapidly induced PASK expression, which peaks 3 days post-injury, a temporal point during regeneration when activated myoblast population is building (Figure 3) [13]. Subsequently, PASK expression declines precipitously once embryonic myosin emerges and regenerative myotube formation is underway around Day 5 [13]. Thus, PASK is expressed at high levels for about 120 h in rapidly proliferating myoblasts during regeneration. It is within this short temporal window during regeneration that the functions and regulation of PASK have been extensively studied.

## 5. Activation of PASK by a Nutrient-Signaling Pathway in MuSCs

The increase in PASK expression in regenerating muscles coincides with the influx of macrophages in the niche. Through the action of macrophages, fibroblasts, and other muscle resident cells, the regenerating niche is enriched with signaling peptides, hormones, and nutrients that guide temporal transition through cell fate and establish a heterogeneous myoblast progenitor population. During regeneration, PASK is activated by insulin and amino acids such as leucine via direct phosphorylation by the mechanistic Target of Rapamycin (mTOR) protein kinase [26]. mTOR plays a multifaceted role in regulating MuSCs activation, proliferation, and differentiation. A portion of PASK is found in association with mTOR complex 1 (mTORC1) under steady-state conditions but is dissociated from the mTORC1 complex upon nutrient stimulation [26]. This is almost certainly due to the multisite phosphorylation of PASK by mTORC1. mTORC1 phosphorylation of PASK results in a modest but consistent activation of its catalytic activity above its constitutive basal levels [26]. Interestingly, mTOR-PASK signaling is specifically required to generate the differentiation-committed progenitor population. PASK is entirely dispensable for myogenesis once nascent myotubes begin to form. That is not surprising since PASK expression is progressively lost after the onset of myogenesis. However, continued activity of mTOR is required to sustain myogenesis program, which is mediated via ribosomal S6 Kinase (p70S6K). Thus, PASK functions in the mTORC1 pathway revealed a signaling dichotomy that underscores the regenerative myogenesis program: PASK-mTORC1 signaling establishes differentiation commitment (via Myogenin, see below), whereas myotube elongation, remodeling and hypertrophy are under the control of the mTORC1-p70S6K pathway (Figure 4).

Mechanistically, the mTOR phosphorylation of PASK at the onset of differentiation strengthens the interaction between PASK and its substrate, Wdr5 [26]. We discovered Wdr5 as PASK binding partner and its substrate by analyzing published high-throughput protein–protein interactions, which were scored against the presence of minimal consensus sequence motif targeted by PASK for phosphorylation [12].

As described below, Wdr5 phosphorylation is a requirement for the induction of the Myogenin promoter and induction of the myogenesis program in response to insulin signaling.

## 6. The Role of PASK in Epigenetic Activation of the Myogenin Promoter

The earliest evidence that PASK is required for induction of Myogenin (Myog) transcription came when we recapitulated a classic MyoD transdifferentiation assay in C3H/10T1/2 cells [13]. In this system, inhibition of PASK nearly completely abrogated the ability of MyoD to induce the myogenesis program in 10T1/2 cells [13]. Since MyoD transduced cells failed to induce MyoG expression when PASK was inhibited, we hypothesized that PASK functions downstream of MyoD activity to regulate the Myogenin transcription. In cultured myoblasts, PASK was required for H3K4me3 deposition at the MyoG promoter upon induction of the myogenesis program. For these functions, phosphorylation of Wdr5 by PASK appears essential since the unphosphorylatable Wdr5 mutant, S49A, failed to induce H3K4me3 modification and MyoD recruitment onto the MyoG promoter. Similarly, stable cell lines expressing Wdr5S49A also failed to induce the myogenesis program downstream of the insulin signaling pathway. In contrast, Wdr5S49E precociously boosted MyoD occupancy at the MyoG promoter and stimulated H3K4me3 modification and myogenesis program (Figure 5) [13]. Thus, PASK signaling to Wdr5 plays is essential for epigenetic activation of the MyoG promoter in response to nutrient signaling. It remains to be determined if there are additional transcriptional targets of pWdr5 downstream of mTOR-PASK signaling that mediate the onset of differentiation.

## 7. Mitochondrial Control of PASK Subcellular Distribution

PASK is primarily localized into the cytoplasm of undifferentiated cells. Interestingly, in freshly isolated primary myoblasts, we noticed that the cytoplasmic localization of PASK is restricted to cytoplasmic granules. However, at the onset of terminal differentiation, PASK is released from these granules and is translocated into the nucleus [19]. As Wdr5 is a nuclear protein, these results suggest that subcellular partitioning is one mechanism to temporally control the onset of the differentiation program. Signaling pathways could target the nucleo-cytoplasmic shuttling mechanism of PASK to induce its nuclear translocation at the onset of differentiation to promote its interaction with Wdr5 [13]. Surprisingly, mTOR-dependent phosphorylation of PASK was not involved in driving PASK nuclear translocation. Instead, mitochondrial glutamine metabolism stimulated p300-dependent acetylation on PASK to drive its nuclear translocation [19]. As glutamine metabolism could also stimulate mTOR activity, these results suggest a multifaceted role of mitochondrial glutamine metabolism in stimulating PASK activation, nuclear translocation, and Wdr5 association. As mitochondrial metabolism switches from fatty acids in QSC muscle stem cells to glucose and glutamine during activation, our results suggest a mechanistic connection between a rewiring of mitochondrial metabolism and the myogenesis program, at least in part, via PASK nuclear translocation (Figure 6).

## 8. Uncoupling of Cell Proliferation from Stemness Machinery by Glutamine-PASK Signaling

In an experimental model of muscle regeneration where chemical or freeze injury to the myofiber network is induced, near-complete myofiber destruction releases MuSCs from their muscle niche, resulting in their activation. Quiescent MuSCs utilize fatty acids, perform beta-oxidation, and cycle infrequently [27,28]. Activated MuSCs consume glucose to generate biosynthetic precursors for nucleotides, cell membranes, and amino acids for rapid growth and proliferation. Injured muscle niche recruit and activate resident and circulatory macrophages, facilitating clearances of damaged myofiber proteins. In addition, macrophages secrete growth factors, bioactive peptides, and nutrients into the muscle niche. Recently, elegant evidence was provided for the involvement of glutamine released from macrophages in controlling MuSCs proliferation and myogenesis [29]. Glutamine is a critical amino acid for rapidly proliferating cells. In addition, glutamine provides carbons and nitrogen for biosynthetic precursors, redox balance, and ATP production. While the role of glutamine in muscle stem cell activation, proliferation, and myogenesis is well-established, the mechanistic details remain unclear.

Mitochondrial metabolism of glucose and glutamine generates co-factors of epigenetic enzymes, such as alpha-ketoglutarate, succinate, and acetyl-CoA, which play critical roles in establishing stem cell fate [30,31]. Intrigued by the observation that the nuclear localization of PASK was powered by glutamine metabolism, we wondered if mitochondrial glutamine metabolism also plays a signaling role in controlling stem cell fate. We first observed that glutamine stimulated PASK-Wdr5 association in muscle stem cells. A mechanistic puzzle started to be revealed when we observed that glutamine withdrawal inhibited the proliferation of MuSCs, but stimulated the preservation of the stemness program, as marked by increased expression of Pax7 [19]. As activated MuSCs begin to differentiate, they generate a heterogenous progenitor population consisting of varying levels of MyoD, Myf5, and Pax7 in individual cells. Glutamine-withdrawn cells, on the other hand, showed near complete loss of heterogeneity due to elevated levels of Pax7 expression in individual cells. Because glutamine depletion halted cell proliferation while increasing stemness, we hypothesize that a cellular program that links the cell cycle with a self-renewal network might be a target of glutamine metabolism in establishing a heterogenous progenitor population.

Self-renewal is a stem cell proliferation feature that preserves stem cell identity and differentiation competence. Thus, the stem cell cycle is uniquely poised to re-establish the stem cell identity at the end of each mitotic cycle. In one mechanism for re-establishing cell identity at the end of the mitotic cell cycle, Wdr5 was shown to recruit mitotic anaphase-promoting complex/cyclosome (APC/C)-Cdc20 to histones of genes regulating ESC identity [32]. APC/C is an E3-ubiquitin ligase, which poly-ubiquitinates and degrades histones at the cell identity genes. That results in rapid promoter accessibility following mitosis and reactivation of cell identity genes. As Wdr5 is a target of PASK activity and PASK nuclear translocation and interaction with Wdr5 is stimulated by glutamine metabolism, we wondered if PASK counters the Wdr5-APC/C interaction to downregulate the self-renewal in myoblasts and generate differentiation competent progenitors. Consistent with this hypothesis, genetic or pharmacological inhibition of PASK resulted in increased Pax7+ myoblast numbers and increased Pax7 mRNA levels. Furthermore, glutamine withdrawal sequestered PASK into cytoplasmic granules and stimulated Pax7 expression [19]. While the addition of a cell-permeable intermediate of glutamine metabolism, dimethyl-alpha ketoglutarate (dm-alpha KG), rescued the increased Pax7 levels in glutamine-depleted conditions, inhibition of PASK rendered dm-alpha KG ineffective at reversing the increased Pax7+ myoblasts in glutamine depleted conditions. Notably, loss of PASK activity did not affect cell proliferation but increased Pax7 transcription, indicating the possibility that PASK inhibition boosts stemness without affecting myoblast proliferation. This suggests that in cycling stem cells, PASK activity results in the downregulation of Pax7 expression, creating transcriptional heterogeneity in proliferating stem cell populations (Figure 6).

Thus, biochemical disruption of Wdr5-APC/C interaction by PASK is required for building a heterogenous myoblast population and generating differentiation-competent progenitors. PASK expression peaks at 3 days post-injury (activated stem cells, ASC, see Figure 6), at which point the niche is enriched with signaling cues such as glutamine, IGF-1, and cytokines released from the macrophages (Figure 6). Thus, the peak of PASK expression coincides with the temporal window when proliferating myoblasts begin to establish their differentiation competence. Consistent with the functional role of PASK during muscle regeneration, we showed that the whole-body *Pask^KO^* animals showed defective regenerative myogenesis and showed the expansion of Pax7+ myoblast numbers throughout the regeneration [13,19]. In uninjured animals, the muscles of whole-body *Pask^KO^* animals appear indistinguishable from WT animals, and the numbers of Pax7+ MuSCs are not different under normal homeostatic conditions. These points to either developmental compensation of PASK function by other kinases or that PASK may not be required for the developmental myogenesis program. Future studies involving conditional MuSCs-specific loss of PASK will be instructive in answering some of these questions.

## 9. Functions of PASK in Organismal Metabolism, Energy Balance, and Metabolic Syndrome

While PASK mRNA and proteins are expressed at nearly negligible levels in most tissues under normal conditions, pathophysiological conditions such as high-fat diet and diet-induced obesity modestly but consistently induced PASK expression [33,34,35]. As a downstream target of mTOR and insulin signaling pathway, PASK plays an important role in organismal energy balance. For example, whole-body *Pask^KO^* animals were protected against high-fat diet-induced obesity and retained increased insulin sensitivity [36,37]. PASK plays a key role in partitioning the glucose fate in yeast and regulating organismal insulin sensitivity in mammals [37,38,39]. Thus, under a high-fat diet feeding condition, *Pask^KO^* animals showed improved insulin sensitivity, lower hepatic triglyceride levels, and increased muscle oxidative phosphorylation [33,37]. Mechanistically, PASK activates the proteolytic processing of SREBP-1c in response to insulin to generate its nuclear form, which transcriptionally drives the expression of FASN, SCD-1, GPAT, and other enzymes involved in triglyceride biogenesis and transport [37]. In addition to regulating hepatic lipogenesis, PASK was recently shown to activate gluconeogenesis in the liver in response to high-fat diet feeding, thereby contributing to hyperglycemia [40]. It is hypothesized that improved muscle glucose uptake and insulin sensitivity in *Pask^KO^* could be attributed to improved hepatic metabolism. As signaling crosstalk between various metabolic tissues has emerged as an important determinant of organismal energy balance, it remains of great interest if the dynamic changes in PASK expression during tissue regeneration influence organismal metabolism. Thus, PASK extensively reprograms metabolism in the liver to drive insulin resistance and has emerged as a potential target for type II diabetes and obesity treatment [41].

## 10. Conclusions and Future Direction

While PASK expression is restricted to the earliest events of skeletal muscle regeneration, it is expressed at one of the most crucial temporal points when myoblasts heterogeneity and committed progenitor population is building in response to cues from the regenerative niche. Our studies indicate that PASK integrates hormonal, nutrient, and metabolic signals from the niche to establish the differentiation competent progenitor population. These functions of PASK underscore its key role in balancing self-renewal, stemness, and differentiation in embryonic stem cells (ESCs) and adult muscle stem cells. There exists a great interest to preserve and extend the self-renewal potential of adult stem cells in vitro. Our results suggest that PASK inhibition might be an important strategy to achieve that goal. At the same time, the in vivo application for targeted inhibition of PASK might benefit organismal metabolism (see above) and likely boost stem cell function, albeit with the caveat of poor differentiation performance. However, since PASK targets two mechanistically distinct pathways to regulate self-renewal and differentiation, it might be possible to develop strategies that promote self-renewal without affecting differentiation performance.

As with any new signaling pathways, many questions about PASK functions in stem cells remain to be answered. For example, additional substrate targets and transcriptional regulators of PASK expression in activated stem cells and at the end of myogenesis programs remain unknown. Furthermore, as most studies of PASK have been conducted in the context of muscle regeneration, it remains unknown if PASK plays similar functional roles in other adult stem cell systems. Nevertheless, emerging studies suggest that PASK plays important role in regulating muscle stem cell function and metabolism, and further studies are needed to fully understand the molecular mechanisms by which PASK regulates muscle stem cell differentiation and energy metabolism and to investigate the potential therapeutic applications of PASK modulation in muscle disorders and aging.

## Figures and Tables

**Figure 1 cells-12-01751-f001:**
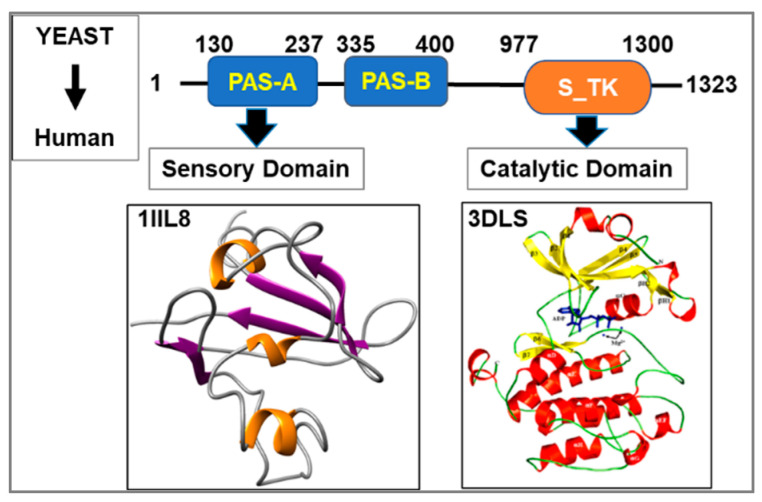
Domain architecture of human PASK. PASK consists of two PAS domains at the N-terminal half of the protein. The catalytic domain is a serine-threonine kinase domain in the CAMK family of proteins. NMR (PAS-A) and X-ray crystal (Catalytic domain) structures have been solved, extensively characterized, and summarized in this review.

**Figure 2 cells-12-01751-f002:**
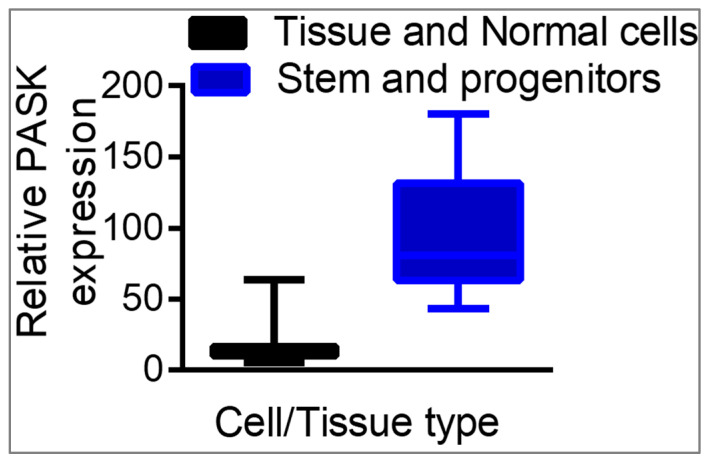
Relative expression of mouse PASK in tissues vs. stem cells. mRNA levels of mouse PASK from differentiated cells or tissues compared to stem and progenitor populations.

**Figure 3 cells-12-01751-f003:**
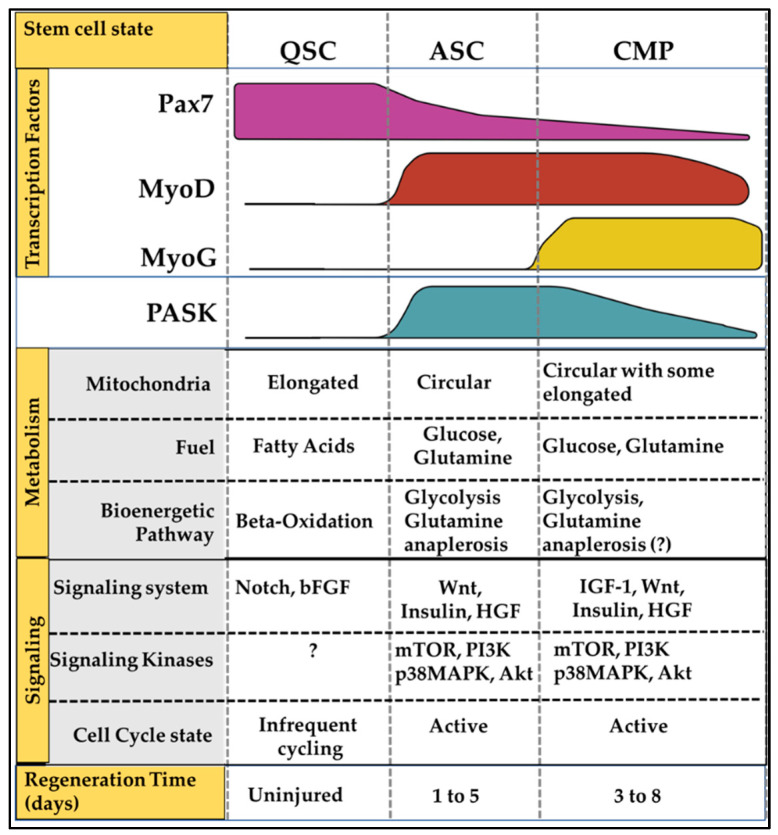
Metabolic adaptations in MuSCs along myogenic progression during regenerative myogenesis. Quiescent MuSCs (QSC), found in uninjured muscles, express Pax7 transcription factor and are dependent upon fatty acid oxidation for bioenergetic needs. QSC state is maintained by Notch signaling at the myofiber–MuSCs junction. Mitochondria in QSCs shows extensive elongated network, which undergoes fission (fragmentation) to generate circular morphology seen in activated MuSCs (ASCs). ASCs express MyoD and switch metabolic fuel preference from fatty acids to glucose and glutamine for rapid burst of proliferation. These metabolic changes accompany increased signaling through mitogenic signaling pathways and increased PASK expression. Transcriptional activation of MyoG marks the Committed Myoblast Progenitor (CMP) state. MyoG positive cells initiate the terminal differentiation program stimulating myoblast fusion program.

**Figure 4 cells-12-01751-f004:**
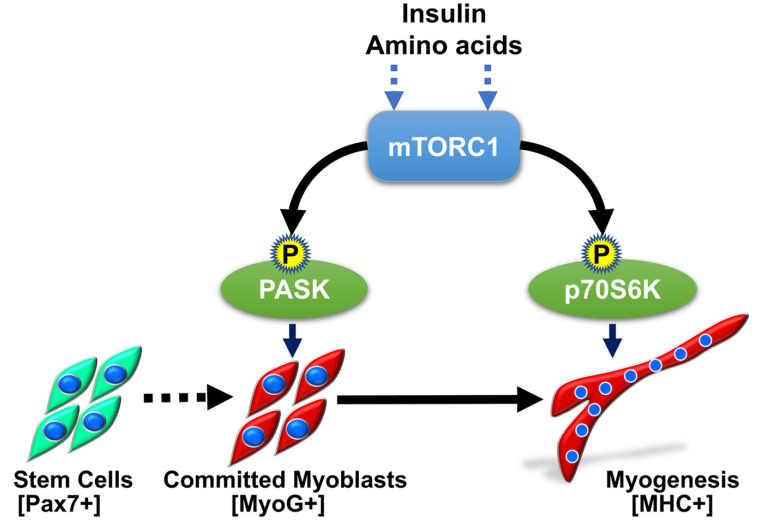
Role of mTOR-PASK signaling in establishment of committed myoblast progenitors. Activation PASK by mTORC1 results in transcriptional induction of MyoG promoter. The subsequent myoblast fusion program is shepherd by another mTORC1 substrate, p70S6K. See text for detail.

**Figure 5 cells-12-01751-f005:**
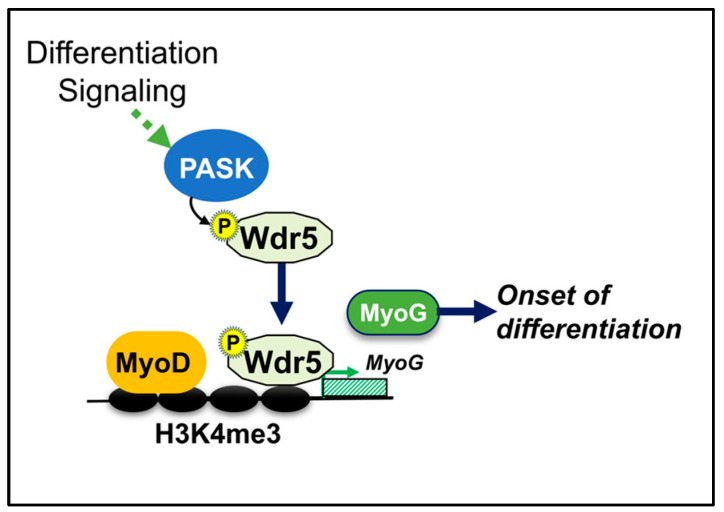
Model depicting the signaling mechanism for epigenetic activation of the *MyoG* promoter downstream of PASK. See text for detail.

**Figure 6 cells-12-01751-f006:**
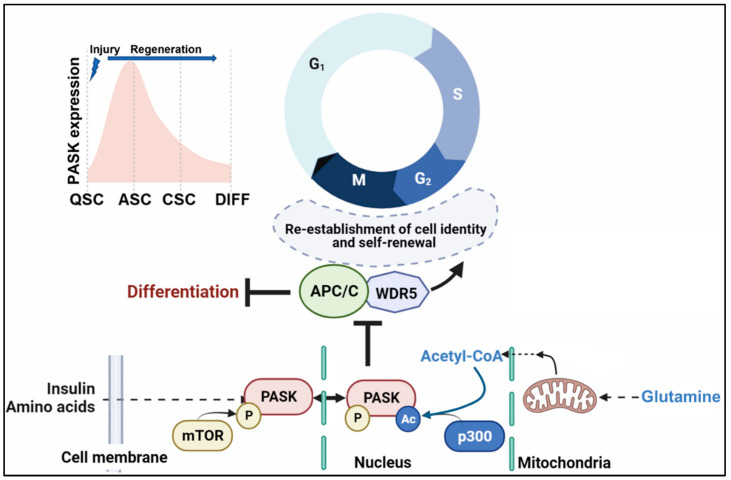
Convergence of mitochondrial metabolism and nutrient signaling on PASK establishes differentiation competence by countering mitotic self-renewal network. Mitochondrial glutamine metabolism boosts stem cell proliferation, PASK acetylation (via p300), and nuclear retention. mTOR phosphorylation of PASK stimulates its catalytic activity and Wdr5 association. Together, nuclear PASK inhibits the interaction between Wdr5 and Anaphase promoting complex (APC) during the G2-M phase of the cell cycle, resulting in the downregulation of self-renewal-related genes and induction of the differentiation program. Acute transcriptional regulation of PASK (see inset on the top left), its subcellular distribution, and mTOR-mediated phosphorylation serve to layer on additional control mechanisms to balance self-renewal and differentiation programs downstream of PASK precisely. QSC, quiescent stem cells; ASC, Activated stem cells; CMP, Committed myoblast progenitors; DIFF, differentiating myoblasts.

## Data Availability

Not applicable.

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
