# Peer review of "Metabolic “Sense Relay” in Stem Cells: A Short But Impactful Life of PAS Kinase Balancing Stem Cell Fates"

_cells, 2023, doi:10.3390/cells12131751_

Round 1

Reviewer 1 Report

This is a very well written and clear picture of the role of PASK in muscle regeneration and stem cell function. The author has presented a nice synthesis of the literature and makes a strong case for the importance of PASK in muscle regeneration and liver metabolism as well.

What is not addressed (but should only need a few sentences) is whether localized PASK activation during muscle regeneration remains localized, or does repair of muscle injury (with a PASK role) also create a systemic impact (i.e., does a muscle injury inducing PASK and then PASK withdrawal affect liver metabolism, gluconeogenesis, lipogenesis etc.). Does PASK regulation in the muscle impact glucose or lipid metabolism in the muscle (i.e., does it affect mitochondria function or have a role in regulating mitochondria biogenesis that would then impact metabolism) also (as is the case in the liver)? A few sentences to discuss whether PASK function remains localized in muscle injury or also has a systemic effect when initiated from a muscle injury, would strengthen the paper.

In the future directions section, the author suggests limiting PASK might be a good strategy to increase muscle stemness, which seems reasonable. However, this statement should be couched in the context of too much of a suppression in PASK might limit muscle regeneration after injury or surgery etc., although being benifical .Thus, there might be a tight balance to regulate PASK. This is an interesting paper, and presents important new work.

Author Response

Dear Editors and Reviewer -1 

Thank you for your effort in evaluating our comprehensive review on PASK signaling in stem cells. As described below, we provide a point-by-point summary of your comments, suggestions, and questions. 

This is a very well written and clear picture of the role of PASK in muscle regeneration and stem cell function. The author has presented a nice synthesis of the literature and makes a strong case for the importance of PASK in muscle regeneration and liver metabolism as well.

Thank you for your generously positive feedback on our work. 

What is not addressed (but should only need a few sentences) is whether localized PASK activation during muscle regeneration remains localized, or does repair of muscle injury (with a PASK role) also create a systemic impact (i.e., does a muscle injury inducing PASK and then PASK withdrawal affect liver metabolism, gluconeogenesis, lipogenesis etc.). Does PASK regulation in the muscle impact glucose or lipid metabolism in the muscle (i.e., does it affect mitochondria function or have a role in regulating mitochondria biogenesis that would then impact metabolism) also (as is the case in the liver)? A few sentences to discuss whether PASK function remains localized in muscle injury or also has a systemic effect when initiated from a muscle injury, would strengthen the paper.

This is a fantastic question dealing with inter- and intra-organelle communication during and following tissue regeneration. Under high-fat diet (HFD) conditions, there are hints of PASK function in the liver impacting muscle insulin resistance. For example, increased hepatic triglyceride output in HFD is linked with muscle insulin resistance, which is ameliorated in PASK-KO animals. We have added a few additional discussion points to address that. 

In the future directions section, the author suggests limiting PASK might be a good strategy to increase muscle stemness, which seems reasonable. However, this statement should be couched in the context of too much of a suppression in PASK might limit muscle regeneration after injury or surgery etc., although being beneficial . Thus, there might be a tight balance in regulating PASK. This is an interesting paper, and presents important new work.

Another excellent point and one that we are actively pursuing. We have clarified that PASK inhibition is focused on the in vitro application to maintain adult stem cell stemness and not in vivo, where it could interfere with homeostatic muscle stem cell functions. Below we directly quote our elaboration of the future discussion point as suggested. 

"There exists a great interest to preserve and extend the self-renewal potential of adult stem cells in vitro. Our results suggest that PASK inhibition might be an important strategy to achieve that goal. At the same time, the in vivo application for targetted inhibition of PASK might benefit organismal metabolism and likely boost stem cell function, albeit with the caveat of poor differentiation performance. However, since PASK targets two mechanistically distinct pathways to regulate self-renewal (Figure 6) and differentiation (Figure 5), it might be possible to develop strategies that promote self-renewal without affecting differentiation performance."

In summary, we express our sincere gratitude for the peer-review effort and many important and insightful comments and suggestions that helped improve the quality of our work. 

Reviewer 2 Report

A few comments to improve the manuscript.

1. Due to differences between human and mouse ESCs/iPSCs, it is necessary to specify species when the author discusses PASK in pluripotent stem cells (e.g., human ESCs, human iPSCs, mouse ESCs, and mouse iPSCs).

2. PASK inhibition preserves ESC self-renewal (lines 119-120). Why is PASK a target of OCT4?

3. In Figure 6, it must clearly label cytosol and nucleus on the bottom panel. Mitochondria are in the cytosol. However, mitochondria could be in the nucleus based on the current panel. 

Author Response

Dear Editors and Reviewer-2

Thank you for your suggestions to improve our manuscript further. We address your comments below and in the revised manuscript. 

1. Due to differences between human and mouse ESCs/iPSCs, it is necessary to specify species when the author discusses PASK in pluripotent stem cells (e.g., human ESCs, human iPSCs, mouse ESCs, and mouse iPSCs).

We have clarified this point in the revised manuscript. As most of our work concerns mouse ESCs or iPSCs at this point, we have emphasized the functions of PASK in that system. 

2. PASK inhibition preserves ESC self-renewal (lines 119-120). Why is PASK a target of OCT4?

This is an interesting and important question. Our model suggests that Oct4-dependent transcriptional activation of PASK prepares stem cells to "sense and relay" signaling cues to start differentiation. As stem cells proliferate, they establish heterogeneously differentiating cell mass across the signaling gradient. Our results show that the inhibition of PASK prevents differentiation and enhances self-renewal using the mechanism described in the review. Thus, Oct4 "pre-loads" PASK to prime stem cells to receive differentiation signaling cues for eventual differentiation. 

3. In Figure 6, it must clearly label cytosol and nucleus on the bottom panel. Mitochondria are in the cytosol. However, mitochondria could be in the nucleus based on the current panel. 

We agree with the suggestion. We have updated Figure 6. Thank you, for the suggestion. 

Kindly yours, 

Chintan

Round 2

Reviewer 2 Report

The author addressed my comments.

Author Response

Thank you!